# Molecular Pathogenesis and the Possible Role of Mitochondrial Heteroplasmy in Thoracic Aortic Aneurysm

**DOI:** 10.3390/life11121395

**Published:** 2021-12-13

**Authors:** A. V. Suslov, M. A. Afanasyev, P. A. Degtyarev, P. V. Chumachenko, M. Bagheri Ekta, V. N. Sukhorukov, V. A. Khotina, S.-F. Yet, I. A. Sobenin, A. Yu Postnov

**Affiliations:** 1National Medical Research Center of Cardiology, Moscow 121552, Russia; suslov_a_v@staff.sechenov.ru (A.V.S.); am-mma@mail.ru (M.A.A.); chumach7234@mail.ru (P.V.C.); igor.sobenin@gmail.com (I.A.S.); anton-5@mail.ru (A.Y.P.); 2Department of Human Anatomy, First Moscow State Medical University (Sechenov University), Moscow 119435, Russia; p.degtyarev@mail.ru; 3Research Institute of Human Morphology, Moscow 117418, Russia; ms.bvgheri@gmail.com (M.B.E.); nafany905@gmail.com (V.A.K.); 4Institute of General Pathology and Pathophysiology, Moscow 125315, Russia; 5Institute of Cellular and System Medicine, National Health Research Institutes, 35 Keyan Road, Zhunan Town 35053, Taiwan; yetshawfang@gmail.com

**Keywords:** thoracic aortic aneurysm, mitochondrial genome, inflammation, metabolism, aortic dissection, cardiovascular diseases

## Abstract

Thoracic aortic aneurysm (TAA) is a life-threatening condition associated with high mortality, in which the aortic wall is deformed due to congenital or age-associated pathological changes. The mechanisms of TAA development remain to be studied in detail, and are the subject of active research. In this review, we describe the morphological changes of the aortic wall in TAA. We outline the genetic disorders associated with aortic enlargement and discuss the potential role of mitochondrial pathology, in particular mitochondrial DNA heteroplasmy, in the disease pathogenesis.

## 1. Introduction

Cardiovascular diseases are among the main causes of morbidity worldwide. Acute and chronic aortic pathologies affect working-age adults, and are characterized by a high mortality rate, therefore being a substantial burden to public health. Several sources have reported that an aortic aneurysm was found in 1–2% of autopsy observations, and its prevalence was increasing with age, reaching a maximum of 10% in the elderly [1]. According to one of the existing definitions, an aneurysm is a local pathological dilation of a blood vessel. A ruptured aneurysm can lead to internal bleeding, frequently with fatal consequences. Aortic aneurysms can be classified based on localization, size and morphological form [2]. Morphologically, aneurysms are distinguished by their shape wall structure. The so-called true aneurysm involves the pathologically altered trilaminar structure of the arterial wall (consisting of intima, media and adventitia). This type of aneurysm may be congenital, atherosclerotic or syphilitic by origin. By contrast, a false aneurysm is represented by small ruptures of the artery that are contained by the surrounding tissues [1].

Abdominal aortic aneurysm (AAA) is a more common type of aortic aneurysm than thoracic aortic aneurysm (TAA). Approximately 5% among men older than 65 years of age have AAA [1]. The morphological features of the thoracic and abdominal aorta can explain the differences between the AAA and TAA pathogenesis. The thoracic aorta is a derivative of the neuroectoderm, while the abdominal aorta has mesodermal origin. Thus, ascending aortic aneurysm is primarily associated with cystic degeneration of the middle layer of the blood vessel (tunica media)—a process that can occur as an integral element of the body’s natural aging (beginning in infancy, but usually manifesting itself after 40 years of age). This process can be accelerated by several factors, such as hypertension, certain genetic mutations leading to connective tissue dysplasia, a bicuspid aortic valve and others. By contrast, AAA is most often caused by atherosclerotic changes in the vascular wall, although the mechanisms of aneurism formation in this section of the aorta remain to be investigated in detail [2,3]. The most dangerous consequences of aortic aneurysm development are dissection and rupture of the vascular wall, which can result in the rapid death of the patient if timely and adequate treatment is not available.

Aortic dissection is customarily classified into acute and chronic, depending on the duration of the disease course [4]. Acute aortic dissection is most common in the ascending section in the thoracic region. A series of autopsy cases revealed the following predisposing factors of acute dissection: bicuspid aortic valve, arterial hypertension and Marfan syndrome [2]. Chronic aortic dissection is more common in the elderly, and is the cause of aortic regurgitation and progressive aneurysm expansion [5].

Although relatively rarely, inflammatory changes in the aortic wall can be the cause of both aneurysm formation and its dissection. Inflammation in the aortic wall is most often caused by bacterial infections, including syphilis. Inflammation of the aorta (aortitis) is detected in the ascending thoracic aorta and can have various nosologic forms: Takayasu aortitis, idiopathic and giant cell aortitis [6]. At the same time, it was noticed that mycotic aneurysms (caused by fungi) most often involve the descending thoracic aorta. Cases of “isolated aortitis”, clinically manifested as independent disorder rather than part of a systemic disease, became increasingly frequent during the recent years. Since aortic aneurysm often does not have clinical symptoms until rupture, early detection and targeted adequate treatment play a crucial role in preserving the patient’s life. Open surgical interventions and/or endovascular (minimally invasive) methods for the restoration of aortic aneurysms are currently generally accepted standard options for the management of such patients. The role of a clinical pathologist in assessing new technologies for treatment of these vascular pathologies is difficult to overestimate.

## 2. Aortic Aneurysm Etiology and Pathogenesis

It should be noted that age-related changes in the aorta are manifested mainly by decreased aortic elasticity. This occurs due to elastin fragmentation with a simultaneous intensification of collagen remodeling, leading to an increase in the collagen/elastin ratio and a reduction of tissue elasticity (extensibility) [7]. Such age-related changes in the aorta inevitably affect its functions. Degradation of elastin causes an increase of arterial stiffness and the speed of pulse wave velocity [8]. Tropoelastin synthesis rate progressively decreases with each decade of life down to 50% by the age of 60, as compared to 20 years of age [9]. Changes of the aortic wall elastic properties are manifested as an increase of tortuosity (deformation of the vascular wall) due to an increase of its inner surface area [9]. It was shown that the greatest expansion occurs in the ascending aorta, and the smallest in the abdominal aorta [10]. At the same time, it was revealed that the thickness of the aortic wall middle layer does not change significantly during a lifetime, while the thickness of the intima increases with age, primarily in the abdominal aorta. This highlights the significant contribution of arterial hypertension and atherosclerosis to the increase of the abdominal aorta diameter and the thickness of its inner wall.

Currently, thoracic aortic aneurysms are registered annually in 6–10 per 100 thousand people, and in 60% of cases, the aneurysms are localized in the ascending aorta [11,12]. Half of patients with thoracic aortic aneurysm have no symptoms (arterial hypertension). These patients are diagnosed using radiological or tomographic methods [13]. Several pathologies, such as annul aortic ectasia, pseudoaneurysm, aortitis (infectious or non-infectious), hereditary connective tissue disorders, atherosclerosis and Valsalva sinus aneurysm led to the formation of thoracic aorta aneurysm. In these cases, morphological changes are described as cystic medial necrosis [14]. A study conducted at Mayo Clinic (USA) examined 513 resections of the ascending thoracic aorta. According to the results of microscopic examination, cystic medial necrosis was detected in 41% of cases. In other cases, aortitis, aortic dissection, a bicuspid aortic valve, hereditary connective tissue dysplasia and normal aortic wall were found [2].

Annul aortic ectasia with cystic medial necrosis, which occurs 2 times more often among males than among females, is characterized by weakening of the aortic root, i.e., all structures of the aortic valve, including fibrous ring, sinotubular crest and Valsalva sinuses. As a result, an extension (aneurysm) is formed. Aortic root expansion can also be associated with developmental abnormalities: coarctation of the aorta and bicuspid aortic valve [15].

Aneurysm of Valsalva sinus is usually a congenital pathology, which occurs four times more often in males than in females [16]. The disease is frequently associated with a ventricular septal defect, a bicuspid aortic valve or coarctation of the aorta. The pathogenesis is linked to abnormal weakness of the sinus wall, leading in turn to the expansion and/or formation of a blind sac (diverticulum) in the wall of one of the aortic sinuses. Acquired Valsalva sinus aneurysm can be associated with infectious endocarditis, and less commonly, syphilis. The simultaneous expansion of all three sinuses (described by the clinical term “aortic root aneurysm”) is usually caused by genetic diseases affecting the connective tissue (Marfan syndrome) [17]. This pathology is especially dangerous because of its asymptomatic course until the aneurysm rupture, which can occur in the right or left heart, pericardium, pulmonary trunk or superior vena cava (depending on the location of the aneurysm). Both aneurysms of aortic arch and the descending part of the aorta are formed as a result of dissection and atherosclerosis of the vessel wall, infectious aortitis, degeneration of the middle layer or Marfan syndrome. Atherosclerotic lesions of the arch and the descending thoracic aorta are found in 90% of cases, while in the ascending part only in 1% of cases [18].

Marfan syndrome was used as a model for studying the pathogenesis of thoracic aortic aneurysm due to a “typical” clinical manifestation of this nosological form. It is a genetically determined disease with an autosomal dominant mode of inheritance caused by mutations in the FBN1 gene (in 75% of cases, these mutations are inherited), which encodes fibrillin-1, a protein located in the extracellular medium [19]. Fibrillin-1 is an important component of elastic fibers, and mutations in the FBN1 gene can lead to a reduction of elastin quantity down to its complete loss. Fibrillin-1 regulates the expression of TGFβ receptors type II [20,21,22]. In addition to Marfan syndrome, there are cases of familial thoracic aortic aneurysm. The genes responsible for this pathology can be grouped according to their functions, including regulation of extracellular matrix proteins (FBN1, COL3A1, EFEMP2); participation in TGFß synthetic processes (TGFB2, TGFBR1, TGFBR2 and SMAD3); and the involvement of smooth muscle myocytes (ACTA2, MYH11, MYLK and PRKG1). The vascular type of Ehlers-Danlos syndrome is caused by genes encoding collagen (COL3A1) [23]. Therefore, part of the aortic aneurysm cases occurs in patients that bear mutations affecting structure proteins of the aortic wall.

One of the serious complications of aneurysms is dissection—a pathology observed with a frequency of about 2 to 4 cases per 100 thousand people per year [24]. It was also determined that aortic dissection occurs 2–3 times more often in males than in females in the absence of Marfan syndrome, and that ¾ of all aneurysm dissections occurs in the ascending part of the aorta. The clinical picture is so unspecific that in every fifth case the correct diagnosis is established only through surgical or post-mortem intervention [24]. Thoracic aortic aneurysm, in comparison with the abdominal aortic aneurysm, is usually an acute process with a rapid death from short-term massive bleeding [25]. The role of genetic factors affecting the connective tissue components is well-established in thoracic aortic aneurysm development.

## 3. The Role of Mitochondrial DNA Heteroplasmy in Pathology Development

The existence of mictochondrial genome was first described in 1963, and human mitochondrial DNA (mtDNA) was sequenced as early as 1981. Human mtDNA is a double-stranded circular DNA consistent of 16,569 nucleotide base pairs, which contains 37 genes, most of which are involved in energy production in the mitochondrial respiratory chain.

Mutations in the mtDNA can occur both in somatic and in germ line cells. Mutations in somatic cells lead to decreased energy production, while mutations in germ line cells can be transferred to offspring, leading to heredicatry mitochondrial disorders. Mitochondrial genome is inherited by maternal lineage. Due to the existence of multiple copies of mtDNA, mtDNA mutations can be homoplasmic, when only one variant is present, such as in case of heredictary mutations, or heteroplasmic, when two or more variants are present [26]. Recent studies using modern sequencing methods demonstrated that somatic heteroplasmy is common in humans [27]. Heteroplasmic oocytes can also transfer mtDNA heteroplasmy for a certain mutation (maternal heteroplasmy) [28].

Mitochondrial genome is less protected against damage and mutagenesis than nuclear genome. Nuclear DNA is protected and stabilized by histone packaging, while mtDNA, although associated with certain proteins, does not benefit from such protection, which can partially explain the higher mutagenesis rate. However, a more important source of mtDNA mutagenesis appear to be the inferior mechanisms of mtDNA replication and repair, as compared to nuclear DNA [29,30]. Mitochondrial mutations include point mutations, deletions and insertions. Moreover, changes in mtDNA copy number also play a role. Mutated mtDNA can be transferred during cell division, together with native mtDNA [31].

Heteroplasmy is a common feature of a group of human pathologies associated with mitochondrial dysfunction. Non-Mendelian inheritance pattern of such disorders makes them challenging to predict [32,33]. Mitochondrial pathology development is defined by the proportion of mutated mtDNA copies (the heteroplasmy level). Normal mtDNA copies can compensate the presence of dysfunctional, mutated mtDNA. At a certain heteroplasmy level, however, such compensation becomes insufficient, leading to pathology development. Therefore, the heteroplasmy threshold can be defined for certain pathological mtDNA mutations [34,35]. Such thresholds can be different for different mtDNA mutations and can also depend on the affected tissue, but are usually found within a range of 60–80% [26,36]. In this paper, we discuss the general information about a certain threshold of mitochondrial mutations, above which diseases develop. In aortic aneurysm, there is also a certain threshold, which is being actively investigated.

## 4. The Mitochondrion at the Crossroads of Metabolism, Inflammation and Immune Response

Human diseases associated with metabolic changes, such as morbid obesity, diabetes mellitus or cardiovascular disorders are frequently linked to chronic inflammation. Pro-inflammatory activation is involved in the pathology development alongside the metabolic changes. The mitochondria, being the cellular powerhouse and the site of vital metabolic reactions, are indispensable for maintaining metabolic and energy balance. Correspondingly, mitochondrial dysfunction is a frequent component of the pathology of metabolic disorders. At the same time, mitochondria play an important role in the inflammatory activation. In immune cells, such as macrophages, mitochondria regulate activation in response to various stimuli, such as pathogen invasion or tissue damage. Mitochondria are highly dynamic organelles that undergo cycles of fission and fusion and are being transported and degraded depending on cellular energy needs. Cells with low proliferation rate and low energy production through oxidative phospholylation control mitochondrial fission and fusion. In the immune cells, mitochondrial population and morphology can reflect the cellular status, which is dependent on metabolic activity. For instance, alternatively-activated (anti-inflammatory, or M2) macrophages usually have elongated mitochondria, and activated T-lymphocytes are characterized by fragmented mitochondria and T memory cells-fused mitochondria [37,38,39].

Therefore, mitochondrial dynamic corresponds to the tissue functional state, being adapted to the current energy needs and providing for an important regulatory mechanism. Such mechanisms appear to be especially important in immune and inflammatory processes. Accumulating data confirm that mitochondria play a fundamental role in immune response development [40].

Mitochondria are surrounded by a double membrane and are semi-autonomous, having their own genome, transcription apparatus and proteome. The outer mitochondrial membrane contains several proteins, such as mitochondrial voltage-dependent anion-selective channel (VDAC), also known as mitochondrial porin [41], mitochondrial antiviral signaling protein (MAVS), which can recognize viruses [42], regulators of mitochondrial fission, such as mitofusin1 and 2 (Mfn-1 and 2) [43,44] and anti-apoptotic proteins such as B-cell lymphoma 2 (Bcl-2) [45,46]. It was shown that VDAC plays an important role in cellular necrosis and apoptosis, regulating the outer membrane permeability [41]. Therefore, the outer mitochondrial membrane can be regarded as a communication channel that connects the mitochondrion with other cellular structures. The inner mitochondrial membrane is not permeable for most ions and molecules, and contains the components of electron transport chain necessary for mitochondrial energy production. The mitochondrial matrix surrounded by the inner mitochondrial membrane has a viscous consistency, and contains mtDNA, ribosomes and soluble enzymes [47].

Therefore, the mitochondrion, due to its involvement in apoptotic and necrotic pro-cesses and regulation of cellular metabolism, can regulate pro-inflammatory reactions in response to tissue damage. Mitochondria also participate in pyroptosis, an inflammatory type of cell death, which is dependent on inflammasome activation. Inflammasomes are multiprotein complexes containing one or more molecule of nucleotide-binding oligomerization domain-like receptors pyrin domain containing 3 (NLRP), apopto-sis-associated speck-like protein containing CARD (ASC) and procaspases. Inflammasomes can recognize pathogen-associated molecular patterns (PAMP) and damage-associated molecular patterns (DAMP)—molecular patterns that are released during pathogen invasion or tissue damage. Among known DAMPs, free mtDNA released upon mitochondrial damage, is known. Activation of PRRs in response to DAMP leads to inflammatory mediator cascade activation, which includes cytokines, chemokines and ad-hesion molecules.

Some molecules of mitochondrial origin, such as *N*-formyl peptide, cardiolipin, mtDNA and reactive oxygen species (ROS) can be regarded as mitochondrial DAMPs that can be recognized by PRRs after their released from the affected tissues by the immune cells. It was demonstrated that mtDNA can induce inflammasome oligomerization and activation in macrophages and T-cells. This leads to activation of interleukin (IL)-1β and IL-18, with subsequent pyroptosis [44,48,49]. Inflammation activation mediated by DAMPs is especially interesting in the context of human pathology. Several studies have demonstrated that inflammasome inflammatory signaling is linked to mitochondrial dysfunction. Mitochondrial damage was shown to induce inflammasome assembly and NLRP3 activation [50,51,52,53]. Therefore, the mitochondrion is not only an energy-producing station of the cell, but also an important inflammatory regulator, controlling both metabolic and immune reactions.

Mitochondrial dysfunction is associated with impaired energy production, which can be caused by mtDNA mutations (mtDNA heteroplasmy) affecting proteins necessary for oxidative phosphorylation or their synthesis [54]. However, mitochondrial dysfunction is characterized not only by impaired oxidative phosphorylation, but also by altered membrane potential, reduction of mitochondrial count per cell and changes of the redox balance due to the elevation of ROS in cells and tissues [55,56].

The respiratory chain necessary for mitochondrial ATP synthesis also serves as a source of mitochondrial ROS (mtROS). For instance, NADH ubiquinone oxireductase (Complex I) and cytochrome c reductase (Complex III) are major ROS sources within the respiratory chain [57,58]. It was shown that mtROS promoted LPS-mediated production of pro-inflammatory cytokines IL-1β, IL-6 and TNF-α, while inhibition of mtROS with MitoQ suppressed p38MAPK activation and IL-6 and TNF-α production [59]. It was also shown that mtROS play an important role in inflammasome activation, which in turn induces activation of pro-inflammatory caspases 1 and 12 and cytokines IL-1β and IL-18 in macrophages [60]. NLRP3 inflammasome appears to be the most sensitive to mtROS. It was shown that NLRP3 interacts with a redox-sensitive protein, which binds thioredoxin (Trx), Trx binding protein-2 (TBP-2, also known as vitamin D3). Elevated ROS mediate TBP-2 dissociation from Trx, therefore promoting its association with NLRP3 and subsequent inflammasome activation and pro-inflammatory signaling [61,62]. Therefore, mtROS generation can be regarded as a mechanism mediating structural and functional changes in mitochondrial dysfunction. Mitochondrial dysfunction in generally accompanied by increased ROS generation with or without oxidative stress development, highlighting the importance of ROS for modulation of intra- and extracellular processes.

Mitochondrial heteroplasmy associated with mitochondrial dysfunction is a cause of mtDAMP and mtROS generation. Presence of extracellular mtDAMPs and mtROS in tis-sues triggers the inflammatory reaction. Recent studies have confirmed the involvement of mitochondrial dysfunction in chronic inflammation [63,64,65].

## 5. Molecular Mechanisms of Fibrogenesis and Mitochondrial Dysfunction

The inflammatory response induced by mitochondrial dysfunction is characterized by the release of inflammatory mediators, such as chemokines, cytokines and adhesion molecules. Recent studies have demonstrated the leading role of chemokines in fibro-genesis activation through chemokine binding to specific receptors in pro-inflammatory cells. Chemokine system includes about 50 ligands and 20 chemokine receptors, many of which regulate fibrogenesis in organs and tissues. Tissue damage results in the activation of a chemokine receptor CCL2, which is involved in organ-independent fibrosis. Studies in experimental animal models demonstrated increased CCL2 synthesis upon fibroblast activation. Experimental models have confirmed the efficacy of using CCL2 as therapeutic target to reduce fibrogenesis [66,67,68]. Certain cytokines can activate specific intracellular signaling pathways of fibrogenesis. For instance, inhibiting of TGF-β was shown to completely block experimental fibrogenesis in the liver, kidneys, heart and skin. TGF-β was shown to play the key role in controlling collagen expression in all organs [69,70,71]. Cytokines of the PDGF family are potent mitogens that regulate cell proliferation and differentiation. It was found that PDGF cytokines are involved in cardiac fibrogenesis. Cytokines stimulate fibroblast migration and proliferation, promote novel fibroblast formation from precursor cells and induce collagen synthesis during disease progression [72,73]. Interleukins IL-1β, IL-6 and IL-33 are known as major inflammatory mediators. Moreover, IL-1β and IL-6 also regulate fibrogenesis [74]. A recent study has shown that IL-33 can promote lung fibrosis, causing imbalance of MMP-9 and TIMP-1 [75]. Another study has demonstrated that IL-33 signal transmission protects cardiovascular system from fibrosis development through controlling fibroblast function and gene expression [76].

As previously mentioned, inflammatory response induced by mitochondrial dysfunction is characterized by mtROS generation. At the same time, ROS generation is a critical factor in fibrogenesis [77]. Currently, a link between ROS and TGF-β is established, with ROS generation increasing the response to TGF-β1. Therefore, increased ROS pro-duction leads to proteolytic activation and increased expression of TGF-β1 in different organs and tissues. One of the key enzymes linking ROS and TGF-β1 is NOX [78]. A study in mice showed that animals deficient for NOX-1 or NOX-4 were protected from fibrosis and liver inflammation [79]. Similar results were observed in kidneys, where decreased NOX-1 and NOX-2 resulted in reduction of trauma-induced fibrosis [80]. Inhibition of NOX-1 in human dermal fibroblasts led to a reduction of TGF-β-induced expression of pro-fibrotic genes [81]. Therefore, ROS generation is a critical factor in fibrogenesis regulation. Generation of mtROS associated with mitochondrial dysfunction can be regarded as an important factor in organ fibrogenesis, alongside inflammatory mediators. Moreover, recent studies from several groups demonstrated the development of oxidative stress in TAA of genetic origin [82,83].

Currently available data indicate that extracellular matrix (ECM) is produced by different cell types during fibgorenesis. Fibroblasts are the most common cell type of meso-dermal origin that are involved in fibrogenesis. These cells produce large amounts of ECM structure proteins, adhesion proteins and connective tissue components. Moreover, they play a major role in maintaining ECM protein homeostasis [84]. Fibrocytes were first described in 1994 as a novel subpopulation of leukocytes. These cells are able to migrate from the blood flow to the site of inflammation, and can be found in scar areas. It was shown that in response to TGF-β, fibrocytes become profibrogenic cells, and not only start to produce ECM components but also take part in angiogenesis [85,86]. Possible precursor cells of myofibroblasts have been identified in recent studies. Such cells (profibrogenic cells) are positive for smooth muscle α-actin (α-SMA) expression, and can produce fibrous collagen and other ECM proteins [87]. It was demonstrated as early as in 1992 that endothelial cells can express α-SMA in response to TGF-β [88]. Such transformation of endothelial cells was termed endothelial-to-mesenchymal transition (EndMT). Since then, numerous studies have demonstrated the involvement of EndMT in fibrosis development in cardiovascular and respiratory systems [89,90]. Vascular smooth muscle cells express α-SMA, which allows considering them as profibrogenic cells. Upon tissue damage, activated profibrogenic cells migrate to the inflammation site and express various ECM proteins [91]. It was shown that in response to bradykinin, vascular smooth muscle cells can synthesize type I collagen [92]. Therefore, excessive accumulation of ECM components during inflammation can affect the stable three-dimensional structure of the organ, leading to changes of its functional status.

Fibroblast transition to myofibroblasts leads to enhanced fibrogenesis. Myofibroblasts and other profibrogenic cells are hypersensitive to fibrogenesis mediators, particularly chemokines and cytokines [93,94]. Therefore, under chronic stimulation with profibrogenic mediators and mtROS, chronic activation of fibroblasts and other profibrogenic cells can lead to ECM restructuring and organ remodeling.

The link between immune and fibrogenic cells is a topic of active research. It is known that immune cells can regulate the full spectrum of fibroblast functional activity. In their turn, fibroblasts can direct the migration of immune cells to the site of inflammation [95,96,97].

Thoracic aortic remodeling is the result of structural and functional changes in the vascular wall, occurring predominantly between endothelial cells (EC) and vascular smooth muscle cells (VSMC). Two mechanisms can be distinguished from the basis of vascular remodeling in an aortic aneurysm. The first mechanism is the degradation of the extracellular collagen matrix, accompanied by the weakened aortic wall. The second one is inflammatory infiltration in the aortic wall, which in turn accelerates the aneurysm progression [98,99,100,101]. The molecular mechanisms of aneurysm development, which arise during mitochondrial dysfunction, considered in this article, are confirmed. In the article, we discuss exosomes containing mitochondrial DNA. Exosomes act as mediators that regulate interactions between intercellular and tissue processes, in particular during physical activity [102]. The article [103] directly reviews intercellular communication mediated by exosomes during vascular remodeling, including aneurysms. A number of recent studies also demonstrate evidence of the involvement of exosomes in the development of vascular aneurysms [104,105,106,107].

## 6. The Double Role of Mitochondrial Dysfunction in TAA: A Unifying Hypothesis

The available published data allows defining the mitochondrial functional state as one of the key elements in regulation of cellular homeostasis. Tight control of mitochondrial function is indispensable for normal cellular activity. Loss of this control (at genetic level) leads to mitochondrial dysfunction, which can be followed by the development of various systemic disorders. For instance, various studies revealed pathological significance of mitochondrial dysfunction for ischemic heart disease [108,109], diabetic cardiomyopathy [110,111], neurodegenerative diseases, obesity [112,113] and others.

Mitochondrial dysfunction leads to the release of mitochondrial components, such as *N*-formyl peptide, mtDNA, mtROS and others, that serve as mitochondrial DAMPs that can activate immune cells. Moreover, such release of mtDAMPs and mtROS can be chronic, leading to chronic inflammation. In turn, chronic inflammation can be regarded as a prerequisite of tissue destruction and organ remodeling.

Inflammation regulates all aspects of tissue repair after activation of immuno-inflammatory reactions, including fibrotic changes and ECM restructuring [114,115,116]. The alteration of ECM structure is linked to loss of connection between smooth muscle cells and elastic fibers. This early event in the pathogenesis of AAT is followed by the reduction of contractile proteins and activation of metalloproteinases, leading to further ECM degradation [108,117,118]. Collagen formation has a protective role in limiting the inflammatory process and subsequent wound healing [119]. However, prolonged changes of ECM structure can lead to tissue remodeling and organ dysfunction. The duration and intensity of the inflammatory reaction determine the process of tissue repair. Failure to resolve the inflammatory process can lead to the persistence of damaging signals, prolongation of the inflammatory response and tissue degradation.

Organ remodeling occurs due to deregulation of the inflammatory process, with mitochondrial dysfunction being one of possible causes. This process can have a key role in TAA formation. In chronic inflammation, myofibroblasts and fibroblasts are hyperactivated (i.e., produce excessive amounts of ECM). Changes of ECM spatial structure leads to aortic wall remodeling, which in turn leads to TAA development.

Therefore, based on the available data, a unifying hypothesis can be formulated, ac-cording to which mitochondrial dysfunction can have a double role in TAA development. This hypothesis is based on the role of mitochondrial heteroplasmy, leading to mitochondrial dysfunction and aortic wall remodeling. Attaining heteroplasmy threshold of mitochondrial dysfunction can therefore cause morphological changes of the aortic wall, contributing to TAA development. These pathological changes can be based on two pathological processes. First, mitochondria can activate pro-inflammatory reaction through mtDAMP release, followed by chronic inflammation and changes of ECM structure. Second, mtROS generation can play a key role in fibrogenesis. Persistent supply of mtROS in the inflammation site can also lead to ECM structure alterations and contribute to chronic inflammation. The described mechanisms can be synergetic, therefore enhancing the pathological changes and leading to TAA development (Figure 1).

## Figures and Tables

**Figure 1 life-11-01395-f001:**
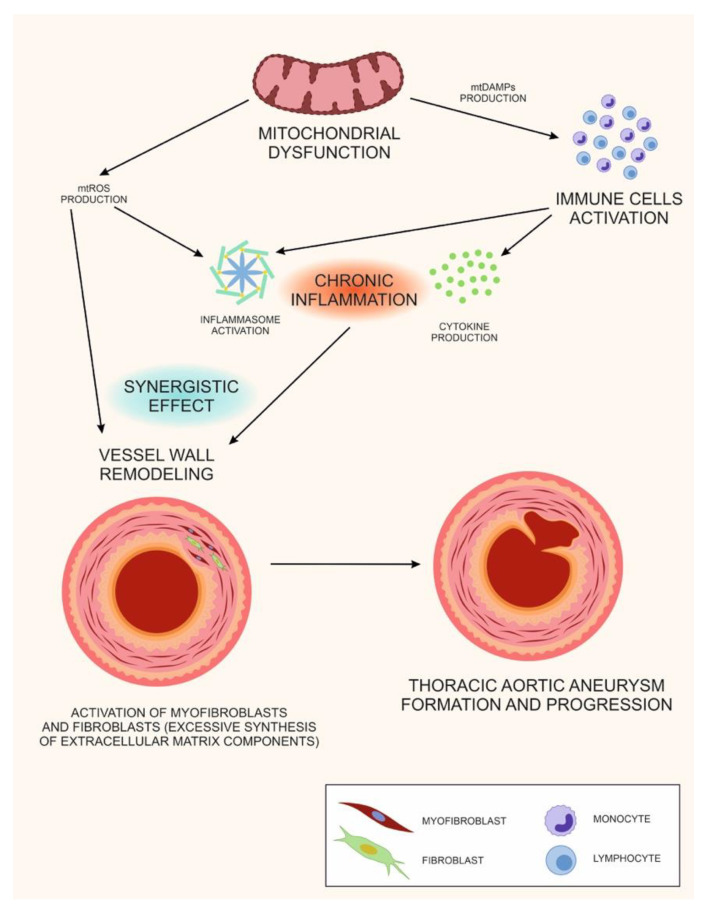
According to this hypothesis, TAA can be placed on the growing list of human disorders associated with mitochondrial dysfunction. During mitochondrial dysfunction, mitochondrial structures (N-formyl peptide, cardiolens, mitochondrial DNA, etc.) are released into the cytosol and onto the cell surface, which act as mitochondrial MtDAMPs markers. MtDAMPs markers can activate the immune cells. Moreover, with mitochondrial dysfunction, MtDAMPs are formed chronically. Thus, the cause of chronic inflammation is mitochondrial dysfunction. During chronic inflammation, myofibroblasts and fibroblasts are excessively active in the tissue (i.e., large amounts of ECM components are synthesized). Violation of the spatial structure of the ECM, leads to remodeling of the aortic wall. As a result of aortic wall remodeling, a TAA aneurysm develops.

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
