# Peer review of "Molecular Pathogenesis and the Possible Role of Mitochondrial Heteroplasmy in Thoracic Aortic Aneurysm"

_life, 2021, doi:10.3390/life11121395_

Round 1
Reviewer 1 Report
The submitted manuscript is interesting and a good review of the present knowledge on the development of thoracic aortic aneurysm. I would change the title to molecular pathogenesis and possible role of mitochondrial DNA heteroplasmy, as there is not much about the morphology in this work. With the title of morphology we expect more data and figures on morphological changes.
To study of exosomes is in the focus of research, to find out their role in the development of pathological changes. What is the role of exosomes in the pathomechanism of aneurism, do we know something about it? It would be interesting to read about it and summarize the present knowledge. (mtDNA content of exosomes and their possible role)
The authors describe that the treshold of heteroplasmy can be defined for certain pathological mtDNA mutations, different mtDNA mutations, depend on the affected tissue, it is usually found within a range of 60-80%. You should explain it in more details in the case of aneurism according to the title of the submitted work.
Author Response
We thank the Reviewer for the critical evaluation of our work! We tried to implement the comments as fully as we could. Please find below our point-by-point response.
Major comments:
- I would change the title to molecular pathogenesis and the possible role of mitochondrial DNA heteroplasmy, as there is not much about the morphology in this work.
- What is the role of exosomes in the pathomechanism of aneurism, do we know something about it? It would be interesting to read about it and summarize the present knowledge.
- The authors describe that the treshold of heteroplasmy can be defined for certain pathological mtDNA mutations, different mtDNA mutations, depend on the affected tissue, it is usually found within a range of 60-80%. You should explain it in more details in the case of aneurism according to the title of the submitted work.
- We changed the name of title to «Molecular pathogenesis and the possible role of mitochondrial heteroplasmy in thoracic aortic aneurysm»
- We added information about exosomes pathomechanism of aneurism. In the article, we discuss exosomes containing mitochondrial DNA. Exosomes act as mediators that regulate interactions between intercellular and tissue processes, in particular, during physical activity. A number of recent studies also demonstrate evidence of the involvement of exosomes in the development of vascular aneurysms. Particularly, authors of “Emerging role of exosome-mediated intercellular communication in vascular remodeling” Su S.A. et al
directly reviews intercellular communication mediated by exosomes during vascular remodeling, including aneurysms.
- Thank you for pointing this out. In this paper, we discuss the general information about a certain threshold of mitochondrial mutations, above which diseases develop.
Reviewer 2 Report
The author described the relationship between mitochondrial DNA heteroplasmy and development of thoracic aortic aneurysm. The amount of information is so huge, and it is quite difficult to understand the paper at a glance. Moreover, after Line 362, the contents of the paper suddenly skipped to the aortic wall remodeling, and it is difficult to understand. Please explain how the contents of Line 1 to Line 362 connect to the aortic wall remodeling more clearly so that everyone can understand your paper.
Author Response
We thank the Reviewer for the time and effort invested in the critical evaluation of our work. The Reviewer’s comments were helpful and allowed improving the manuscript considerably. We implemented them as fully as we could. Please see our point-by-point response below.
- After Line 362, the contents of the paper suddenly skipped to the aortic wall remodeling, and it is difficult to understand. Please explain how the contents of Line 1 to Line 362 connect to the aortic wall remodeling more clearly so that everyone can understand your paper.
We appreciate the reviewer’s feedback, so we have added the paragraph to connect Line 1 to Line 362. Two mechanisms can be distinguished from the basis of vascular remodeling in an aortic aneurysm. The first mechanism is the degradation of the extracellular collagen matrix, accompanied by the weakened aortic wall. The second one is inflammatory infiltration in the aortic wall, which in turn accelerates the aneurysm progression. The molecular mechanisms of aneurysm development, which arise during mitochondrial dysfunction, considered in this article, are confirmed.
Round 2
Reviewer 1 Report
You have involved a new author to the manuscript, why was it necessary?
I accept your answers, but you could write more about heteroplasmy.
Author Response
We thank the Reviewer for the critical evaluation of our work.
We have involved Dr.Pavel A Degtyarev because he was erroneously omitted during the submission of our manuscript. We noticed our mistake only during the revision. We claim that he took part in the preparation and the revision of our paper.
We thank the reviewer’s suggestion to write more about heteroplasmy. It would have been interesting to explore this aspect. However, we decided to discuss mitochondrial heteroplasmy in brief because we thoroughly reviewed the heteroplasmy in our previous papers (DOI: 10.1155/2014/292017 ; DOI: 10.2174/1381612811319330013 ; DOI: 10.3390/biom9090455).
Reviewer 2 Report
Thank you for the revision.
Author Response
Thank you.